# Responsible Innovation in Business: Perceptions, Evaluation Practices and Lessons Learnt

**Agata Gurzawska**

Department of Philosophy, Faculty of Behavioural Sciences, University of Twente, P.O. Box 217, 7500 AE Enschede, The Netherlands; a.m.gurzawska@utwente.com

**Abstract:** This study derives from the results of the European Union (EU)-funded SATORI (Stakeholders Acting Together on the ethical impact assessment of Research and Innovation) project. It seeks to gain insights about, firstly, integration of the responsible innovation (RI) concept into companies' practices; and secondly, various evaluation approaches to companies' innovation practices that consider responsibility, ethics and sustainability. Twenty four interviews with companies and business experts were conducted to understand the ways in which companies apply principles, frameworks and evaluation practices related to RI. The results emphasize the confined character of companies' RI practices in the context of corporate social responsibility (CSR), sustainability and ethics. Moreover, the results indicate two main types of RI evaluation and control among companies, namely assessment and guidance. This paper discusses theoretical and practical implications of discrepancies in understanding and evaluating RI for large corporations and small and medium-sized enterprises (SMEs). Consequently, new approaches to RI in business are proposed, calling for strategic and responsible innovation management.

**Keywords:** responsible innovation; business practices; CSR; ethics; sustainability; ethics assessment; guidance

## 1. Introduction

Industry plays a crucial role in European innovation as the main funder of European innovation and the principal agent of novel technological solutions [1]. The unburdened promotion of innovation supports economic growth and creativity. Nevertheless, real life cases, such as Volkswagen's emission scandal, misuse of data by Cambridge Analytica or the Hacking Team's surveillance services targeting human rights activists and journalists, raise legitimate concerns abut whether science and technology can be left to operate autonomously in the market without regulation and societal guidance.

Policy-makers and social scientists have introduced the concept of responsible research and innovation (RRI) to counter this and encourage innovation that is ethically acceptable and socially desirable [2]. Since the vast majority of research and innovation is funded and produced by industry, a growing body of research focuses on the implementation of RI in industry [3–10]. Nevertheless, companies tend to have virtually no awareness or recognition of this concept [7,11]. The discourse on the implementation of RRI in the business context has evolved into a link with the more widely known notion of corporate social responsibility (CSR) and corporate sustainability (CS). As a result, for the business context specifically, the simpler term "responsible innovation" (RI) has emerged, which has been used synonymously with the abbreviation "RRI" [12].

One of the major drawbacks to adopting RI is a lack of unity, of recognised approaches and professional standards for implementation and evaluation of RI [13]. Ethics assessment is a key element of RRI, which enables the identification and assessment of ethical issues in research and innovation [13]. Companies evaluate and control their activities and impact, including innovation, as part of their strategies to ensure that they meet the desired outcomes and create value. However, there is still considerable ambiguity with

regard to evaluation of innovation aimed at strengthening companies' responsibility and strategic planning, exploring competitive opportunities and mitigating negative human rights, societal, ethical and environmental impacts [14]. Furthermore, the private sector is diverse, perceptions and approaches developed and applied by large companies may not be well-suited for small and medium-sized enterprises (SMEs), and vice versa.

This study investigates, first, how companies perceive and integrate the RI concept; and second, how companies evaluate their innovation practices by considering responsibility, ethics and sustainability. Twenty four interviews with companies and business experts were conducted to understand the ways in which principles and practices of RI and evaluation of innovation vary for companies. Consequently, this research aims to determine the extent to which similarities and differences exist in the use of frameworks and procedures. This paper discusses theoretical and practical implications of discrepancies in definitions of responsibility, sustainability and ethics, language used, differences between large corporations and SMEs. As a result, it proposes new approaches to RI in the business context calling for a strategic and responsible innovation management. This study derives from the results of the European Union (EU)-funded SATORI (Stakeholders Acting Together on the ethical impact assessment of Research and Innovation) project [15].

This paper is organised as follows. Section 2 gives a brief overview of the relevant research by introducing the concepts of RRI and RI, and comparing them to the related notions of CSR and CS. Moreover, this section explains the role of evaluation and control as necessary condition for responsible innovation in order to strengthen strategic planning and explore opportunities and mitigate risks. Section 3 describes the methodology for data collection and analysis for this study. In Section 4, a descriptive approach is taken to present the results of the interviews with companies. Deriving from the analysis of companies' experiences, the paper presents specific approaches of how companies deal with responsible innovation and its evaluation. Based on the empirical findings, Section 5 takes a normative perspective and proposes new approaches to RI in the business context. Lastly, Section 6 summarises the findings.

## 2. Responsible Innovation in Business: Depicting the Field

This section provides an overview of concepts related to responsible innovation in the business context, namely corporate social responsibility (CSR) and corporate sustainability (CS), responsible research and innovation (RRI), and a number of evaluation approaches aiming at assessing companies activities in terms of responsibility, sustainability and ethics, as well as technology and innovation assessment approaches.

### 2.1. Corporate Social Responsibility (CSR) and Corporate Sustainability (CS)

The concept of business responsibility that goes beyond immediate shareholders and making profits has a long history in the business management literature that can be traced back to the 1950s and 1960s. Business responsibility towards society has been conceptualised in various ways, through corporate philanthropy, business ethics, corporate citizenship, stakeholder management, corporate social performance, and recently the most dominant concepts of CSR and CS [16]. The most often cited definition of CSR is Carroll's (1979) who conceptualises CSR as 'the economic, legal, ethical, and discretionary expectations that society has of organisations at a given point in time' [17]. Other definitions emphasise five dimensions of CSR, namely environmental, social, economic, stakeholders and voluntarism [18]. In principle, CSR can be linked to four theories: instrumental, political, integrative, and ethical [19].

Corporate sustainability derives from the concept of sustainable development defined in the Brundtland Report [20]. CS is generally defined in two ways, either as primarily focused on the environmental dimension of business; or in a broader sense, includes environmental, economic, and social dimensions [21]. While we still lack a standardised definitions of CSR and CS [18,19,22], in broad terms they focus on responsibility, hence

duties and obligations or motivation and opportunities of the companies towards the environment and the welfare of society [14,23].

Since the 2000s, there has been a growing interest in the business case for CSR and CS [16]. It has been claimed that social responsibility and sustainability can go hand in hand with economic development creating "shared value" [24]. Gugler and Shi (2009) claim that the economic interests offered by CSR such as better access to market, finance and business; enhanced intangible assets, reputation, community relations; and reduced risk from regulatory sanction, could encourage companies to structural changes including innovative processes and technological upgrading [25]. As a result, the concept and scope of business responsibility towards society has also evolved, from mere philanthropy actions to the so-called strategic CSR where CSR lies at the core of the business model and is brought into central value creation [26–28]. Responsibility dwells in the management of business operations as well as the impacts of their activities on the environment and society. This study perceives corporate responsibility as a business strategy, where responsibility is designed to create business value and positive societal and environmental change and is managed in a systematic and intentional way [29,30]. Therefore, social responsibility is embedded in a day-to-day business culture and operations [31]. However, from a conceptual point of view, CSR and CS tools or actions are generally not designed specifically for innovation. CSR and CS cover all aspects of a company's activity but do not exclusively relate to a company's innovation activities.

### 2.2. Responsible Innovation

Innovation refers to application of new ideas for product, process, organisational and marketing innovations [32], among which some are technological (technology-based new products, processes or features) and others non-technological innovations (social or organisational in nature). Innovation is crucial for companies' profitability and long-term survival, because it enables a company to adapt to the dynamically changing needs of the marketplace [33]. While innovation leads to commercial and financial success [34], it is now increasingly recognised by policy-makers and society that it is important for innovation to be performed responsibly and ethically. The concepts of sustainable innovation, environmental innovation, eco-innovation, open innovation and social innovation are among the most commonly discussed developments in the business context that reflect this change.

The most recent development is the concept of RRI used in EU policy and academic studies to refer to research and innovation that is ethically acceptable and socially desirable, where the science outcomes are aligned with the needs and values of the society [2,35,36]. The aim is to encourage societal actors to work together during the whole research and innovation (R&I) process to better align R&I and its outcomes with the values, needs and expectations of society [37]. Recent works on RRI emphasize various conditions of innovation process, such as a need to include stakeholders [38], a need for diversity and equality for gender, which should also be anticipatory and reflexive [39]. Furthermore, innovation process should be open and transparent, responsive and adaptive to change [38,40,41]. Additionally, the policy-makers highlight the importance of science literacy, science education and open access to scientific knowledge [42].

Studies suggest that so far companies do not recognise the RRI concept [7,11]. There are various reasons and challenges for the implementation of RRI in the business context. RRI is being developed by science policy-makers, various funding agencies (e.g., European Commission) and academia [43,44]. Yet the interests of academic researchers and policy-makers may differ from the interests of innovators in the business context, because commercially driven innovation focuses on the economic impact, as argued by Lubberink et al. (2017) [11]. Therefore, some aspects of RRI may have conflicting aims and trajectories for industry's objectives, such as promotion of science literacy or open access to scientific knowledge and research results (intellectual property) [45,46]. Even more of a challenge is the question of companies' motivation for engaging in RRI [47,48].

Definitions and key constructs for CSR, CS, RI have proliferated during the past decade enhancing uncertainty. Ambiguous definitions and constructs may prevent companies from identifying and implementing RI goals for their companies. Nevertheless, this does not necessarily mean that companies innovate in a irresponsible way. This study responds to the question of how companies perceive responsibility in the context of their innovation practices and how they implement this responsibility. A growing body of literature sheds light on the implementation of RI in business (a term more often used in the business context than RRI), including RI principles [9]; as well as incentives, drivers and barriers of RI [4,45,49]. This study responds to the call for learning from the way RRI is implemented in companies [50], thus it empirically investigates the state-of-the-art of RI approaches in industry contributing to a flourishing research on the implementation of RI in the business context [7,51,52]. Moreover, it confronts policy-makers' and academics' ambitions with the current practices of companies.

### 2.3. Innovation Assessment for Responsibility and Ethics

One can ask: how does one know that someone innovates responsibly? A successful implementation of RI requires anticipating potential ethical, societal and environmental opportunities and challenges, as well as envisioning impacts of the innovation process and outcomes. Evaluation and control are an inevitable element of companies' strategies together with planning and implementation [53]. Therefore, the following questions arise: how can one know if an innovator is responsible if she does not evaluate and control her innovation? How one can identify and evaluate ethical, societal and environmental issues for technologies that are still emerging because they are still at the innovation stage? What standards and assessment methods one should follow to ensure that innovation processes and outcomes are responsible? Who should conduct such assessment and who else should be involved? Having a better understanding of the place of innovation evaluation, for enhancing responsibility, within the family of previously developed assessment forms may help to contextualise the evaluation of RI by companies.

The fields of CSR and CS suggest that companies apply various evaluation approaches to ensure their operations and outcomes are responsible and sustainable. A multidimensional CSR or CS assessment may involve a range of assessment methods such as impact assessment (IA). IA is the process of identifying the future consequences of a current or proposed action [the International Association for Impact Assessment], e.g., effects on environment (environmental IA), on society (social IA), health (HIA) or human rights (HRIA).

Technology and innovation management studies involve a broader field of technology assessment (TA). TA aims to evaluate potential, and actual, impacts of new technologies on industry, the environment and society; and to develop instruments to steer technology development in more desirable directions [54–56]. The assessment is based on known or potential applications of the technology, taking into consideration consequences that are unintended, indirect or delayed [57].

RRI emphasises that innovation should be assessed and evaluated with the goal of influencing innovation processes to make them more ethical. As a result, ethics and ethics assessment (EA) have emerged as a key element of RRI evaluation, involving the identification and assessment of ethical issues in R&I. The SATORI project focuses on EA of R&I, defined as 'any kind of assessment, evaluation, review, appraisal or valuation of R&I that makes use of ethical principles and criteria' [13]. The evaluation criteria of EA are guided by ethical principles to determine whether certain actions or developments are right or wrong, referring to individual and collective rights (e.g., freedom and privacy), benefits and harms (e.g., towards society or environment), fairness and virtues (e.g., integrity) [13]. EA is distinct from other forms of assessment, because it uses normative ethical criteria in assessment. As identified by the SATORI project, EA is increasingly institutionalised and, increasingly, R&I plans, practices and products are subject to ethical review [13].

While TA and IA evaluation methods have a long history, recently new types of evaluation have emerged which incorporate the ethics dimension. Two of the most notable

methods are ethical technology assessment (eTA) and ethical impact assessment (eIA). The first one proposes the engagement of ethicists in technology development throughout the entire lifecycle of development projects to confront developers with ethical issues [57]. The second is an approach that takes into account the specific context and engages stakeholders to find ways of dealing with ethical issues arising from the development of new technologies [58].

All these evaluation approaches share a common intention of facilitating the social shaping of innovation. Taking into consideration a multiplicity of approaches to responsible business and innovation assessment, this study investigates companies' practices for assessment of their innovation processes and outcomes aiming at enhancing responsibility. This study aims to map companies' evaluation and control approaches, applied tools and methods; and verify these instruments to identify good practices and gaps.

## 3. Methodology

This study draws on lessons learnt from the business world, the academic concept of CSR and our experiences in the EU-funded SATORI project about RRI and ethics assessment and ethical guidance in different fields, organisations and countries [59]. This research focuses on innovation activities of the private sector, in a way that reflects the current companies' practices and opinions of business experts. To that end, the empirical component of this paper entails interviews and case study reports. The interviews aim to gather information and opinions from, and about, different companies regarding practices of, and attitudes towards RI, and evaluation of innovation.

In total, 24 interviews were carried out in person and via phone and Skype in one to one and half hourly slots. In addition to the interview data, desk research was employed to compile the case study reports, making use of the academic and non-academic (e.g., ethics codes) literature and material found online (e.g., reports, website descriptions of companies) [14]. The interviewees involved companies (and organisations of companies) and experts in responsible business. The first group includes large corporations, SMEs and organisations of companies from top sectors that engage in research and development (R&D) (pharmaceuticals and biotech, automobiles and parts, electronics and electrical equipment) and the lowest R&D engagement (oil and gas, general industrials) [60] (18 interviews). The interviews involved mainly top level management of research and development (R&D), innovation and CSR/CS personnel. The second group engages persons who were regarded as experts in the area of innovation, responsible business, and human rights and business on their work in the field and knowledge of the research. These persons included specialists from consultancy, academia, research institutes and non-governmental organisations (NGOs) (6 interviews). The experts bring a broader perspective on practices of companies because of their experience in collaborating and providing consultancy to companies of various types and sizes. The goal was to generate a broad overview of responsible innovation practices, because the selected experts often engage in evaluation of companies' activities in terms of responsibility. The sampled companies and experts were selected following the overall project's methodology aimed at mapping and analysing the ethics assessment landscape for R&I in the EU, where countries were used as the main structuring principle for data collection [13]. The sampled companies and experts represent both large companies and SMEs from various parts of EU, featuring different institutional and cultural arrangements. Table 1 presents the sampled informants from companies and organisations of companies.

**Table 1.** Interviewees representing companies.

| Informant | Company's Sector | Major Activity | Size |
| --- | --- | --- | --- |
| 1 | Electronic and Electronic Equipment | Research and development (R&D) | Small and medium-sized enterprises (SME) |
| 2 | Electronic and Electronic Equipment | Manufacturing | Large |
| 3 | Oil and Gas | Energy production | Large |
| 4 | Oil and Gas | Energy production | Large |
| 5 | Oil and Gas | Energy production | Large |
| 6 | Oil and Gas | Energy production | Large |
| 7 | Pharmaceuticals and Biotech | R&D | Large |
| 8 | Pharmaceuticals and Biotech | R&D | SME |
| 9 | Pharmaceuticals and Biotech | R&D | Large |
| 10 | Pharmaceuticals and Biotech | R&D | Large |
| 11 | Pharmaceuticals and Biotech | R&D | Large |
| 12 | Pharmaceuticals and Biotech | R&D | Large |
| 13 | Pharmaceuticals and Biotech | R&D | SME |
| 14 | Automobiles and Parts | Manufacturing | Large |
| 15 | General Industrials | R&D | Large |
| 16 | General Industrials | R&D | Large |
| 17 | Various | Various | Large and SMEs |
| 18 | Various | Various | Large and SMEs |

The interviews were guided by the interview template (Appendix A), but with the flexibility to use any additional relevant questions (including factual ones). The interview template was developed from a literature survey of the scholarly and grey literature on RI in the business context, CSR, SC, RRI, innovation management and assessment [14]. The semi-structured interviews ensured enough leeway to facilitate modification, elaboration and occasional digressions. During the interview, the interviewees were informed of the aim and the use that would be made from the information and opinions provided in the interview. They were informed that no full transcript of the interview would be produced, only a summary. Interviews were only taped with prior permission of the interviewee, and explanation was provided of the use of the tape. Anonymity was assured, unless requested otherwise. If the interviewee had so requested, they were sent a copy of the summary for their comments.

The in-depth interviews and additional documents were then analysed using the coding qualitative interview analysis technique [61,62] and resulted in collected data being coded with a thematic analysis approach [63]. This process was interactive and iterative, involving breaking down, examining, comparing, conceptualising and categorising data. A final edit of the codes revealed the following two main themes with a number of sub-themes: (1) perception of RI; and (2) RI evaluation and control. First theme includes three subthemes: (a) overarching concepts (umbrella) under which interviewees put RI; (b) what specific topics interviewees associate with RI; (c) principles that characterise and guide com-

panies' innovation processes. The second theme includes three subthemes: (1) assessment; (2) guidance; and (3) dissemination and awareness rising. The following section presents the results of the empirical studies providing an overview of the responsible innovation perceptions and innovation assessment practices among companies.

## 4. Results

This section discusses the results of interviews with companies and business experts. The results are divided into two aspects of implementation of RI in the business context. First, I present interviewees' perceptions of responsibility in the context of innovation and the role of RI in companies' strategies by specifying overarching concepts, reoccurring topics and principles which guide companies' innovation processes. Second, I demonstrate evaluation and control practices of companies that aim to determine whether certain innovation processes and outcomes are responsible, ethical and sustainable.

### 4.1. Perception of Responsibility and Responsible Innovation (RI)

Table 2 shows the meanings of concepts by presenting two aspects of RI perceptions: (1) overarching concepts (umbrella) under which interviewees put RI; and (2) which specific topics interviewees associate with RI. The majority of those interviewed recognise companies responsibility towards society and the environment. Interviewees clearly refer to responsibility, responsible behaviour, and corporate responsibility. The analysis of the interviews (Table 2, Overarching concepts) shows that more than two thirds of those interviewed discuss their responsible innovation practices in the context of CSR, and half of them in relation to CS. Interestingly, the interviewees seem to either blend CSR and CS and use them interchangeability, or refer to CS specifically in the context of environment and sustainable use of resources, particularly in the oil and gas and energy related sector. The most striking result to emerge from the data is that only one third of the participants refer to responsible innovation, however framing it as innovating responsibly or, ethical innovation, environmental or eco-innovation and social innovation. Those who refer to some sort of responsible innovation engage in projects funded as part of the EU R&I programme, particularly in topics related to renewable energy solutions or science with, and for, society. Nonetheless, those interviewees who acknowledge the term "responsible innovation", also tend to link it with CSR and CS and conceptualise companies' ethical, societal and environmental responsibilities as part of a broader CSR or CS strategy.

For the vast majority of the interviewees, responsibility applies to all aspects of a company, not exclusively to innovation. RI is, therefore, part of broader policies and strategies for CSR or CS. A CSR or CS policy is intended to function as a self-regulating mechanism for business to ensure its compliance not just with laws, but also with the spirit of the law, with international norms and with ethical standards. At the same time some interviewees feel that bigger companies put a lot of emphasis on CSR whereas smaller companies with a lesser public interface are less likely to do so. Some companies are aware of the need to retain public support and based on that work to improve CSR/ethical practices. A number of interviewees emphasise the role of their corporate social and sustainability responsibility as an implicit part of the company's culture and activities, and therefore their strategy. For instance, one of the interviewees says that: *'Long-term thinking and responsible action are the basis for our business success ( . . . ) Social and environmental responsibility is an integral part of how we perceive ourselves as a company'*. Therefore, responsibility goes into the core of their business. Another interviewee comments on the role of ethical behaviour: *'We are convinced that an ethically correct behaviour can give a positive return also on the bottom line. The cost for respecting ethics is not considered a cost, but an investment which pays back'*. Some interviewees also feel that a good corporate reputation differentiates a company from its competitors. One of the experts emphasises that it is important for enterprises to understand that bad management of corporate responsibility is not only a matter of image, but it exposes the company to high risks, decreasing the total value of the business.

**Table 2.** Responsible innovation overarching concepts and associated topics.

| | Number of Companies | Number of Experts | Total |
|---|---|---|---|
| *1. Overarching concepts* | | | |
| Corporate social responsibility (CSR) | 14 | 4 | 18 |
| Sustainability | 9 | 3 | 12 |
| Responsible innovation (including research, ethical innovation, environmental innovation, social innovation) | 6 | 2 | 8 |
| Innovation detached from responsibility | 2 | 2 | 4 |
| *2. Associated topics* | | | |
| Environmental responsibility | 16 | 4 | 20 |
| Anticipation and reflection (including evaluation, assessment, guidelines) | 14 | 5 | 19 |
| Social responsibility | 14 | 4 | 18 |
| Stakeholders: | 13 | 3 | 16 |
| - Community and society | 5 | 4 | 9 |
| - Employees | 6 | 0 | 6 |
| - Stakeholders (general) | 4 | 2 | 6 |
| - Customers and users (including specific types e.g., patients) | 5 | 0 | 5 |
| - Shareholders | 4 | 0 | 4 |
| - Business partners and supply chain | 2 | 2 | 4 |
| Ethics (including business ethics) | 12 | 4 | 16 |
| Economic responsibility (including profit) | 7 | 2 | 9 |
| Gender equality (including diversity and inclusiveness) | 7 | 1 | 8 |
| Openness and transparency | 6 | 0 | 6 |
| Legal responsibility | 3 | 2 | 5 |
| Governance | 5 | 0 | 5 |
| Science education | 4 | 0 | 4 |
| Voluntarism | 3 | 0 | 3 |
| Public engagement | 1 | 0 | 1 |
| Open access | 0 | 0 | 0 |
| Responsiveness and adaptation to change | 0 | 0 | 0 |

Source: Terms are derived from the most reoccurring terms related to responsible business, corporate social responsibility (CSR), corporate sustainability (CS), responsible innovation (RI) and responsible research and innovation (RRI) (see Section 2). The results are based on the interviewees' responses.

At the same time, four of the participants argue that innovation should not be restricted by any means, especially by responsibility or ethics. For two of the company representatives, the main argument is that for companies innovation is risky and costly. If innovation would become part of the responsible innovation agenda, taking into account also sustainability

and labour rights, this would limit the creativeness of firms. It is the role of the state to solve societal issues, they claim. The role of a company is different in this regard, its role is to make a profit, provide a work place and pay taxes to the state. These interviewees emphasised the principle and strengths of the free market. One of the experts also wonders who should invest in e.g., social innovation, the public or private sector. Another two experts emphasised that the correlation that exists between CSR and innovation is less critical, even in the case of a business with high technological value. In their opinion, the true challenge is to connect CSR with the actual business performance.

Since the vast majority of interviewees recognise their broader responsibility, the next step is to see how interviewees perceive this responsibility. A variety of perspectives are expressed, however five main topics related to responsibility and innovation emerge from the analysis (Table 2, Associated topics), namely:

1.  Environmental responsibility
2.  Anticipation and reflection (including evaluation, assessment, guidelines)
3.  Social responsibility
4.  Stakeholders
5.  Ethics (including business ethics)

A common view amongst interviewees is that environmental responsibility plays a crucial role in their responsible business practices, including innovation. This opinion is shared by companies from all business sectors, not only directly related to the environment such as automotive, electronics and oil and gas, but also pharma or general industrials. The second topic is anticipation & reflection that involves evaluation and assessment of envision impacts. A vast majority of those who were interviewed agree on the third topic in the context of RI, i.e., social responsibility in the sense of an obligation to act for the benefit of society at large. The forth topic is related to responsibility towards stakeholder and stakeholder engagement, including external stakeholders (such as local communities, society at large, customers and users, e.g., patients, business partners and supply chain) and internal stakeholders (mainly employees and shareholders). For instance, one of the interviewed companies holds stakeholder forum involving ministries, sub-companies, automobile clubs, environmental organisations, and universities in order to discuss new developments such hydrogen or shale gas. Lastly, ethical responsibility is recognised as one of the crucial aspects of RI. However it should be noted that interviewees are unanimous in their views about ethics. They refer to ethics as either a general moral framework, or specifically medical ethics, (particularly pharma and biotech interviewees), or business ethics understood as aspects related to anti-corruption and anti-bribery. For pharma companies in particular, ethics seems to be an overarching framework. A number of interviewees claim that ethics and ethical issues are part of the everyday routine related to drug development (e.g., a reference to deontology, patient-centred ethics). The findings indicate that other themes, particularly related to RRI [2,35,36,42], do not seem particularly prominent in the interview data.

Table 3 focuses on guiding principles for companies' behaviour. When the interviewees were asked about principles that characterise RI and that guide their innovation processes, there is a sense of shared crucial principles of RI amongst interviewees (Table 3). The most shared principles involve:

1.  Social responsibility
2.  Environmental impacts
3.  Professional integrity
4.  Implications for health and/or safety.

**Table 3.** Principles of RI behaviour in business.

| RI Behaviour Principles | Number of Companies | Number of Experts | Total |
|---|---|---|---|
| Social responsibility | 13 | 3 | 16 |
| Environmental impacts | 13 | 2 | 15 |
| Professional integrity | 13 | 1 | 14 |
| Implications for health and/or safety | 11 | 2 | 13 |
| Social impacts | 7 | 1 | 8 |
| Equality/non-discrimination (e.g., gender) | 7 | 1 | 8 |
| Human subject research | 5 | 3 | 8 |
| Implications for quality of life | 6 | 1 | 7 |
| Treatment of animals in R&I | 5 | 2 | 7 |
| Scientific integrity | 5 | 1 | 6 |
| Implications for privacy | 5 | 1 | 6 |
| Human dignity | 4 | 1 | 5 |
| Implications for civil rights | 3 | 1 | 4 |
| Justice/fairness | 3 | 0 | 3 |
| Outsourcing of R&I to developing countries with lower ethics standards | 2 | 1 | 3 |
| Autonomy/freedom | 0 | 1 | 1 |
| Dual use (possible military uses) | 0 | 1 | 1 |
| Other, specify: | | | |
| Transparency | 3 | 0 | 3 |
| Human rights | 0 | 2 | 2 |
| Freedom of market | 1 | 1 | 2 |

Source: Based on the interviewees' responses.

The importance that the interviewees give to the first two principles, (i.e., social responsibility and environmental impacts) reflects the topics that recurred throughout the dataset (Table 2, Associated topics), namely attention to the environment and the welfare of society. Third principle, professional integrity, refers to professional standards comprised of practices, ethics, and behaviours that members of a particular profession must adhere to. The fourth shared principle's implications for health and/or safety relates to measures in relation to the employees' safety, security and health, but also related to health and/or the safety of customers and users of their products and services.

Other principles indicated by the interviewees vary depending on the context of innovation and the sector. For instance, in pharma and biotech, higher priority is given to principles related to drug development, experiments and clinical tests, such as scientific integrity, transparency in the publication of results of clinical trials, human subjects research (patients/human safeguard), human dignity, treatment of animal in experiments, privacy, equality and non-discrimination in access to treatment, high ethical standards related to outsourcing of research and/or innovation to developing countries. For automotive, oil and gas and energy-related sectors, environmental and social impacts play a crucial role, but also implications relating to distributive justice, individual and civil rights and quality of life. The electronics sector's interviewees emphasise importance of implications for individual and civil rights and privacy.

The vast majority of participants demonstrate their commitment to responsibility, ethics and sustainability. However, this commitment requires specific actions in terms of planning, implementation and evaluation. Drawing on the perception of RI among companies and business experts, the next step is to understand how companies ensure that their innovation activities live up to their declarations. Therefore, the next section analyses companies' approaches to RI evaluation and control.

### 4.2. RI Evaluation and Control

The results of the interviews presented in the previous section suggest that anticipation and reflection is an important aspect of RI in the business context. Furthermore, the results demonstrate that a vast majority of interviewees perceive RI evaluation as part of, either their CSR and CS frameworks, or risk management.

Overall, evaluation and control of innovation can be generally divided into two main categories (1) assessment; and (2) guidance. Both assessment and guidance are oriented towards evaluation of projects and practices on the one hand, and professional conduct on the other hand. As a result of the interviews, a third main category of evaluation and control emerged, which involves engagement in networks and trainings. This category is called dissemination and awareness raising. The participants on the whole demonstrate that they engage in evaluation of their innovation activities to anticipate and reflect on responsibility. Nearly half of those who were interviewed declare having assessment procedures or guidelines in place, or a combination of both (Table 4). Only a small number of interviewees indicated that they do not evaluate innovation activities in terms of responsibility, ethics or sustainability. Two discrete reasons emerge from this. First, some participants feel that innovation and responsibility are disconnected, and it is the role of the market and consumers to decide whether they want to use a product or service. Second, others consider that currently they do not have such procedures in place, but they admit that such evaluation might be helpful.

Interestingly, there is a difference in the ratios of responses to a general question asking whether the interviewees engage in assessment or guidance, and responses about specific assessment or guidance methods and approaches. While the general responses indicate a lower engagement in RI evaluation practices, the more detailed questions about RI evaluation methods show a higher number of RI evaluation practices. The most likely two explanations of this result are: firstly, the interviewees misunderstand the terms "assessment" and "guidance", thus there is a language difference (especially in the context of the SATORI project that focuses on the ethics assessment of R&I). As emphasised by one of the interviewed experts, RI is not a reference point for them. When they analyse pharma companies developing drugs or activities of electronics companies, they use the CSR or human rights and business frameworks and language. Secondly, the incorporation of RI processes into a more general corporate social and sustainability framework causes confusion about whether particular innovation activities are subjected to evaluation, and if yes, which assessment tools are relevant in this context.

**Table 4.** RI evaluation and control.

|  | *Number of Companies* | *Number of Experts* | *Total* |
|---|---|---|---|
| *1. Evaluation activity* |  |  |  |
| **Assessment** | 8 | 2 | 10 |
| Compliance assessment | 10 | 2 | 12 |
| Impact assessment (IA) | 8 | 2 | 10 |
| Ethics assessment (EA) | 8 | 1 | 9 |
| Safety assessment | 3 | 2 | 5 |
| **Guidance** | 8 | 2 | 10 |
| External guidelines and principles | 15 | 3 | 18 |
| Standards | 10 | 2 | 12 |
| Internal code of conduct | 7 | 2 | 9 |
| **Other** |  |  |  |
| (Dissemination & awareness raising) |  |  |  |
| Network | 11 | 3 | 14 |
| Training | 7 | 2 | 9 |
| **None** | 4 | 1 | 5 |
| *2. Type of evaluation & control* |  |  |  |
| In-house | 14 | 3 | 17 |
| Outsourced | 4 | 2 | 6 |
| Other | 0 | 0 | 0 |

Source: Terms are derived from the SATORI project (see Section 3). The results are based on the interviewees' responses.

The vast majority of evaluation activities are conducted as internal procedures within a company (in-house, 17), as opposed to external review (outsourced, 6). As reported by interviewees from both large companies and SMEs, they rather have no separate units or personnel to evaluate and control their innovation activities. For large companies, most of the internal evaluation are conducted by units dealing with CSR, CS or risk management with oversight from boards of directors; and as stated by one interviewee, all activities must be carried out having in mind their company's code of conduct (self-evaluation). One interviewee illustrates the situation in the following way: '*Strategic importance is given to the centre's sustainability in these times of economic crisis, which means resources are reassigned to core activities (identification of R&D opportunities, identification and development of technology-based business opportunities, internationalisation strategy, technological excellence etc.). Resources assigned to deployment of initiatives, compliance and reporting are scarce. Internal communication about CSR strategy and policy has been scarce. There are several lines of work being developed and not always in close coordination'.* Furthermore, one of the interviewed experts suggests that corporate responsibility units do not have the power to influence the strategy or the organisation structure and assets. More likely, in order to obtain results, whole companies should be re-addressed to be corporate responsibility friendly; otherwise the business paradigm will not work in this sense. This view is shared by another interviewee who emphasises that: '*It is a challenge to keep high the attention of the management and employees on these topics, even in period where there are other urgencies or challenges to be faced and therefore people tend to focus only on core business issues.'* Regarding SMEs, one of the interviewees from an SME emphasises that generally, SMEs do not have enough resources for having people dedicated solely to this task. Sometimes certain problems could be overlooked or

not timely addressed. Nevertheless, the creation of a company vision, with written policy on this matter, could help to overcome this problem. The interviewee added that they are aware of the need of innovation assessment and its impacts, and they try to address timely the relevant issues involved, though without a structured approach.

*4.3. Assessment*

The interviews indicate four assessment methods and approaches that interviewees refer to as relevant for RI practices, namely:

1. compliance assessment;
2. impact assessment (IA);
3. ethics assessment (EA);
4. safety assessment.

The most common assessment procedure that companies implement is compliance with laws and regulations to ensure that regulatory requirements are met (12). The interviewees allude to the notion of assessment of legal or risk compliance.

The second commonly used assessment approach is IA in its numerous forms (10). The types most referred to include environmental, social, human rights and health impact assessments. The majority of interviewees who use IA are large companies. One interviewee states that the goals they set in the IA process are included in a management plan, and they must be implemented and tracked through the life of the project or operation. Furthermore, in terms of stakeholder engagement in the IA, two divergent and often conflicting discourses emerged. Some of the interviewees emphasise the role of a dialogue with stakeholders in the development process to mitigate potential risks and optimise the delivery of benefits. Others note that there is no real need to engage stakeholders or the general public in the impact assessment procedure. For instance, one interviewee from a large electronics company argues that there is no need because the market will verify a company's investments; the interviewee from a large pharma company says that they would not engage the general public but typically consult with medical experts within the field. Interestingly, one of the interviewees reports that their core business is innovation, thus their CSR strategy and deployment methodology and tools to evaluate the impact of projects were developed in collaboration with the innovation strategies department, and externally with association for promotion of social technology, local government and SMEs.

A number of interviewees refer to ethics assessment (EA) as part of their innovation activities (9). There are four discrete contexts in which EA is conducted, firstly interviewees who participate in activities funded by the EU and where ethics appraisal is an integral part; secondly, in regard to pharma companies which engage in drugs development involving human subject research, clinical trials, animal experimentation and require ethics approval from ethics committees; and thirdly, EA in the context of data protection and privacy of individuals. EA in the context of business ethics and professional behaviour, for instance related to bribery, fraud or conflict of interests. Nevertheless, EA seems to be conducted in an administrative or reactive way. One of the interviewees feels that there would be a specific need for future-oriented ethical impact assessment (eIA) that realises the assessments in the context of specifically built future scenarios considering, for example, emerging technologies. On the other hand, one expert suggests that an eIA would fall to a great extent within other types of IA, e.g., social impact assessment (SIA), environmental impact assessment (EIA), health impact assessment (HIA) or human rights impact assessment (HRIA). The eIA should be carried out, but the interviewee would not call it eIA, but rather as one of the commonly known IA. The reason for this is that other IA methods are already recognised by the community as types of IA. Therefore, instead of developing a new type of IA it should be integrated with the existing types. Lastly, safety assessment also plays a role in RI, particularly in the context of to safety and efficacy of a product and safety in the workplace (5).

The empirical findings show that the RI assessment in the business context is conducted both internally (in-house) and externally (outsourced). The internal assessment

is conducted by, for example, CSR/sustainability officers or departments; the external assessment by external/independent auditors, for example, consultancy companies or organisations with expertise in responsible business. Furthermore, some types of assessment are mandatory such as projects in the context of medical innovation involving clinical trials, or projects funded by the EC. Other types of assessment are voluntary. The interviews show that companies use assessment tools, even if they are not mandatory. As emphasised by one of the interviewees, such an assessment is an investment of time and resources, but can strengthen responsible and ethical behaviour and, long term, could help to anticipate eventual future problems. Lastly, RI assessment is either informal or formal. An informal assessment, may have a form of, for example, a reflection discussion within a company's board or CSR department. A formal assessment may take a form of annual reporting. This tool reflects issues of transparency, anti-corruption and tax-avoidance. Nevertheless, non-financial disclosure on the environmental and social impacts is receiving a wider acceptance in the business world. Recently, companies have been becoming more willing to provide this information voluntary as a part of their annual report and by participating in such initiatives as Global Reporting Initiative (GRI). The interviews reflect this tendency, and a number of interviewees representing large corporations declare that they disclose information on environmental and social performance. One of the interviewees points out that the integrated report they publish regularly is designed to describe their system of quality control as well as structure and governance.

An important aspect of evaluation is the criteria that companies use to assess their performance. A number of interviewees emphasise the importance of quantitative indicators that help to measure the actual social, environmental, ethical and sustainability performance, and the impact of their activity. The results of the interviews suggest that one of the main assessment criteria that companies use is key performance indicators (KPIs). KPIs are used to evaluate businesses' success at reaching targets and to demonstrate in a quantifiable way how effectively a company is achieving key business objectives. Two of the interviewees declare that responsibility/or sustainability has been established as a strategic corporate objective based on specific targets and KPIs. As stated by one of the interviewees: *'Sustainability is an explicit component of our management system. This means on the one hand that every major project must be measurable in terms of sustainability as a corporate objective, ensuring that, in addition to economic factors, environmental and social aspects are also accounted for in the decision-making process'*. The interviewed experts agree that such indicators are crucial for understanding the impact that businesses have on our lives, nevertheless they point out that there seem to be a lack of agreement around which KPIs should be taken into consideration, especially in the context of RI. Nevertheless, the interviewees suggest that such set of KPIs should be adapted to the context and not "one size-fits-all". A number of interviewees indicate that the assessment criteria should be installed in day-to-day activities. They suggest that one way of connecting companies' responsibility, ethics and sustainability objectives is to do so through connecting these objectives to management objectives. Employees would be incentivised with bonuses for meeting the targets. One participant comments that evaluation measures have a very positive impact on the company. The evaluation has some organisational costs, but it contributes to improving the quality of the company. These processes are binding and a failure to comply is punishable.

### 4.4. Guidance

The second type of RI evaluation and control practice is standard-setting guidance, which refers to the statement of guidelines, principles, rules, codes, and recommendations to which innovation practices are expected or recommended to adhere [13]. Guidance presents ideals to live up to or norms to follow [13]. Guidance differs from assessment, because it does not involve judgment about a specific project or action: it is not the case that particular types of innovation, or its use in society, are judged to be responsible or irresponsible [13]. Rather, guidance sets general standards according to which any

specific activities or outcomes of innovation may be guided. The results of the interviews indicate that guidance are either developed by a company itself (internal) or by other organisation (external). The interviewees provide examples of such guidelines that lead their companies in their responsibility activities. Guidelines include codes of conduct, international and sectoral guidelines and principles, standards, and reporting initiatives to provide quantitative data on corporate responsibility performances. As one interviewee puts it:

> *'The conviction [belief] that ethical behaviour is not a cost but a profitable investment could promote the adoption of ethical practices, but clear and precise rules could certainly also help. We prefer them, although besides the rules to be followed, we adopt also internal self-regulatory tools that can strengthen an ethical behaviour and in a long-term perspective could help to anticipate eventual future problems'.*

A vast majority of interviewees, particularly representing large corporations, declare that they follow internationally recognised guidelines and principles of responsible business (18), such as the United Nations (UN) Global Compact; the United Nations Guiding Principles on Business and Human Rights (UNGP); International Labour Organisation Tripartite Declaration of Principles concerning Multinational Enterprises on Social Policy; and the Organisation for Economic Cooperation and Development (OECD) Guidelines for Multinational Enterprises. Furthermore, the interviewees indicate that their activities are also guided by sectoral guidelines and principles, such as the Code of Conduct of the Electronic Industry Citizenship Coalition (EICC) and the European Federation of Pharmaceutical Industry Associations (EFPIA). One of the interviewees feels that more companies should agree and subscribe to a common set of principles such as the UN Global Compact in order to ensure more ethical corporate behaviour. Interestingly, the interviewee does not see the need for guidelines specific for the pharma industry because they are to a great extent covered by good clinical practice and the Declaration of Helsinki, but setting minimum standards, a tool-kit or conduct of responsible and ethical research across industries or groups that engage in research could be helpful.

A specific type of guidance are codes of conduct. Around one third of interviewees confirm having such instruments demonstrating their values and commitment that is a good practice guide for employees and business partners in their daily work (9). Interviewees say that codes of ethics ensure the utmost diligence, professionalism, transparency, collaboration, and availability.

Moreover, the results of the interviews show that also standardisation plays a role in companies' RI activity (12), because it provides clear requirements on development and implementation of management strategies. According to the interviewees, companies' commit to implementation of the following standards: ISO 26000 Guidance Standard on Social Responsibility (ISO 26000); Social Accountability 8000 (focusing on workers' rights and workplace conditions); OHSAS 18001 (regarding the health and safety of employees and minimising the risk of accidents); ISO 14001 and Eco-Management and Audit Scheme (EMAS).

*4.5. Dissemination and Awareness Raising*

The last type of RI evaluation and control that emerges from the interviews is dissemination and awareness raising. A recurring theme is the engagement of companies in a broader discussions about general business responsibility towards the environment and society, as well as technological developments and their impact on changing our life, at the international and sectoral level (14). Such engagement takes various forms, such as participation in multi-stakeholder or sectoral initiatives at the international and national levels. The interviewees also emphasise the role of building a culture of responsibility within a company (employees) and its eco-system (business partners, supply chain). A number of interviewed companies engages in awareness raising and building competency thorough trainings (9). They promote employee awareness of company policies; and ensure

safeguards to protect bona fide "whistle-blowing" activities. Figure 1 illustrates the RI evaluation and control practices in the business context.

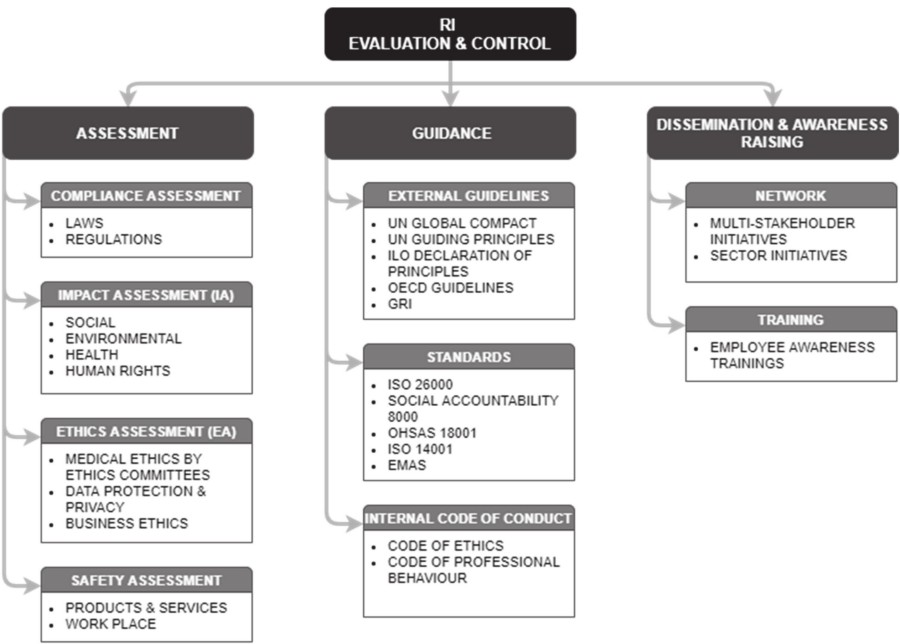

**Figure 1.** Summary of the RI evaluation and control practices in the business context.

## 5. Discussion

The results of the interviews show diversity in RI perceptions, implementation approaches, evaluation and control methods among companies. This section discusses the similarities and differences between the companies, using their characteristics and differences in orientation, innovation processes and cooperation. Furthermore, drawbacks of current state-of-the-art are considered, and alternative approaches that call for a strategic approach to RI in the business context are proposed. The proposed solutions are supported with opinions expressed by the interviewed companies and experts.

### 5.1. RI Concept

The majority of those interviewed recognise companies responsibility towards society and the environment, nevertheless there is no wider recognition of the RI concept among companies [7,11,47]. The interviews clearly show the theoretical confusion regarding the definition of responsibility [6]. The concepts of CSR, CS and ethics are used by interviewees interchangeably and unsystematically. Companies, policy-makers and academia speak different languages [11,43,44]. As a result, the perception of RI by companies is vague, partially convergent (anticipations and reflection, internal and external stakeholders engagement, ethics) but also somehow distinct (reference to specific aspects of responsibility such as environmental impacts, professional integrity, and implications for health and/or safety) from the definitions proposed by the policy makers and social scientists. A clear definition of RI is necessary for creating awareness among managers and innovators of their responsibility towards society [64].

The interviewees show a general commitment to perceiving innovation as an inherent part of their CSR and CS strategies. On the one hand such an approach suggests that responsibility is integrated at every aspect of a company's activity, thus also innovation. This approach follows the view that a company is a system of interrelated and interdependent parts [53], thus responsibility and innovation are connected and have equally important roles. On the other hand, the study shows a scarce interaction and communication between departments. Internally, different departments work separately, not always in a close coordination and cooperation. Current practices of companies do not reflect the complex and

multifaceted reality of modern research and innovation ecosystems [65]. While, literature argues that organisational learning though assimilation of existing knowledge and the generation of new knowledge is crucial for adoption of responsibility within companies [66,67], this study shows that companies often overlook the importance of knowledge management as part of their responsibility [11].

In line with the existing literature [5,68], the interviewees present two conflicting views about the role of innovation, either as interlinked with the corporate responsibility framework, or as oriented mainly towards financial success. This dichotomy triggers a question about the real substance of companies' claims and whether they are "walking-the-talk". The results suggest that RI may share the same path as CSR, which was originally meant to strategically shape the corporate identity of companies, but currently it is criticised for mainly focusing on corporate philanthropy [69,70]. There is a risk of "misusing" RI for marketing purposes by misleading consumers about social, ethical and environmental benefits of a product or service (e.g., greenwashing). The interviewed experts emphasise that it is important for companies to understand that a bad management of corporate responsibility is not only a matter of image, but it exposes a company to high risks [49] and missed business opportunities, decreasing the total value of a company.

The definition of innovation implies that it is a strategic and multidimensional process that affects all units in a company, its organisational structure, people, processes, procedures, and systems [53]. Therefore, it is crucial that innovation is managed in a strategic way ensuring cooperation and communication between technology developers and CSR/CS/ethics people. Therefore, corporate responsibility, including responsibility in the context of innovation, should be tied to business strategies and performance, through a systematic approach involving planning, implementation, evaluation and control [71].

*5.2. RI Evaluation and Control*

Findings of the interviews suggest that evaluation and control of RI fall under companies' CSR and CS practices. The study corroborates previous findings [5,9,72] by demonstrating that RI practices are guided by general CSR and CS guidance frameworks, including standards, global initiatives and assessment tools. Monitoring the performance of companies supports CSR and RRI by identifying areas for improvement and potential drawbacks Companies evaluate and control their innovation practices through assessment, guidance and dissemination and awareness raising. They apply well-known assessment tools, such as those focused on compliance, ethics, impact and safety, helping in decision-making informed evaluation of the economic, social, and environmental effects [65]. Nevertheless, this research shows a discrepancy between the high importance that the interviewees give to principles of environmental and social impacts that should guide the RI processes and outcomes and a relatively low level of the actual use of evaluation methods of such impacts.

The study shows a general confusion around what should be evaluated and controlled; no distinction around methods and tools applied for evaluation of the innovation process and outcome is made. Such a distinction is mainly used in the pharma industry, where innovation processes are subjected to ethical evaluation, covering various areas of research that involves humans or animals (research ethics). This is because a great majority of research ethics is comprehensively regulated at the national, EU and international level (e.g., human embryonic stem cell [hESC] research, clinical trials, children, animals, bioethics, dual use, biosafety). Dreyer et al. (2017) propose that research and innovation processes differ, thus programs, tools, and criteria for defining best practice, as well as governance mechanisms, must also follow different rules and principles [6]. Stahl et al. (2017) identify various components of RI (including purpose, process, and product aspects) and identify five maturity levels (unaware, exploratory, defined, proactive, and strategic) [7]. Moreover, Werker (2020) proposes four major features which are crucial for assessing, namely (1) innovative agents; (2) Innovative agents' communication and collaboration with partners form their relationships; (3) formal and informal institutions; (4) innovative agents' activities [73].

This diversity of evaluation approaches has a strong disadvantage. It leads to a confusion, the principles, standards and initiatives of which are core. Despite this variety of initiatives, the interviews show the absence of strategic CSR or CS tools explicitly devoted to innovation activities that would be integrated within a broader responsibility framework. Under RI, anticipation refers to systematic thinking about emerging critical issues and discovering new possibilities and opportunities [69]. Nevertheless, the results show that companies lack such systemic evaluation practices that would assess the degree to which a company's practices align with RI [7,23] or innovation activities that aim to reduce the uncertainty around potential negative impacts of innovation [11]. Furthermore, companies generally focus on evaluation and mitigation of risks, rather than critically examining which desirable implications are missed by the innovation [11]. The interviewed experts emphasise that general approaches are ineffective. Implementation of RI requires adequate information and mechanisms for companies to take responsibility and for stakeholders to hold companies accountable. This study confirms the need for translating RI into business-relevant KPIs [10,23]. Such indicators are crucial for companies to measure their performance and impact, but also for customers, policy-makers and broader society to understand impacts that companies have on our lives. The RI is context-sensitive [74], thus KPIs should reflect such diversity. Evaluation and reporting frameworks should be multi-layered, providing general principles applicable to all types of actors, as well as specific provisions suitable for different types and categories of actors (e.g., branches of industry). RI evaluation may be used to check compliance [75]. However, the RI reporting, which is a crucial aspect of communication with external stakeholders, should become much more than an occasional press release, it should be a recognised way of companies' evaluation.

Some of the interviewees argue that companies do not need new tools; what they need is the integration of currently existing tools in order to avoid an overlap and provide a clear, fully compatible and flexible responsible business framework. Nevertheless, new and emerging innovations, such as data-driven policing tools, cellular senescence and life extension, or 3D printed molecules, are complex technological developments. Thus, their potential impacts may go beyond the immediately obvious applications. The currently used tools may not be adapted to capturing such complexities. Innovation by definition takes place in the future, thus we need anticipatory approaches to shed light on social and environmental impacts [76]. Nevertheless, the interviews do not show that more technology oriented assessment methods (e.g., eTA or eIA) are use by companies. RI in business requires interdisciplinary cooperation and integration of existing approaches, such as ethics and social sciences, in a novel way by shifting focus and placing new emphases [77]. This approach could be applied through such methods as eTA, eIA, anticipatory technology ethics (ATE), value-sensitive design (VSD), privacy for design, socially responsible design (SRD), eco-design, ethics by design etc. A mutual understanding may enhance responsibility and open new business opportunities.

The benefits of responsibility, ethics and sustainability may not be straightforward, which can easily result in undervaluation of its principles. Responsibility always comes from individual values [45]. A vast majority of companies depend on self-evaluation in daily activities. However, if employees, management, owners of the company, their customers and business partners do not understand or appreciate responsible values and principles, it is difficult to capture the benefits of RI. A company may translate the ability of individuals to understand the impact of RI through statement of a company's principles, goals, strategies that involve responsibility, training and education. Recent research by Meijer and van de Klippe (2020) suggests that the future RI evaluation and monitoring should focus on the institutional change through 'empowering individuals to articulate their own values within their institutions by providing them with intellectual resources to do so' [78]. The elements of ethical leadership are also crucial for incorporating RI leading to good governance and responsible organisation [6]. Furthermore, the interviews reveal an important role of companies' participation in various multi-stakeholder and sectoral initiatives that serve as a forum for learning, sharing and standard setting. Recent develop-

ments in the area of RRI, mainly outcomes of the EU-funded research projects, offer various tools that may support businesses to explore responsible innovation opportunities. One example is the Responsible Innovation COMPASS self-check tool developed with intention to help SMEs determine to what extent their practices align with RI principles, how to improve their innovation processes and outcomes, and how they compare to other companies [79]. The MoRRI (Monitoring the Evolution and Benefits of Responsible Research and Innovation) project developed a list of RRI indicators for adequate measurement of responsibility in research and innovation, which could serve as KPIs [80]. Other initiatives provide lessons-learnt through pilot studies engaging companies [81] and co-creation of good practices through workshops and community networks [82,83].

### 5.3. Large Companies and Small and Medium-Sized Enterprises (SMEs)

Larger companies generally have structured and complex governance structures, the spectrum of actors on which they have an influence and their impacts are likely wider than SMEs [84]. Nevertheless, large companies have complex organisational structures, and therefore responsibility for implementation and oversight require compound implementation and evaluation approaches that address various organisational levels.

Regarding SMEs, according to interviews findings, SMEs generally do not have CSR or CS structured strategies, tools or reporting. This does not mean that SMEs manage their business irresponsibility; however without the evaluation approaches it is difficult to depict trends and behaviours about their positive/negative role in the society [84]. A number of interviewees emphasise that SMEs lack human and financial resources. This situation reduces SMEs' ability to undertake research and development, limits opportunities for commercialisation of innovations [85,86]. These constraints drive the goals of SMEs to be relatively short-term and profit-oriented [86]. Although SMEs seem to be less equipped for RI, their nature can compensate their resource shortcomings, particularly a simple organisational structure, an informal and entrepreneurial leadership style, flexible organisation capacities, better efficiency and responsibility-oriented personnel benefit SMEs over large companies [87–89]. Nevertheless, for both large companies and SMEs commitment from the leadership (the board, chief executive officer (CEO), director of the organisation) is considered paramount. The definition and review of the responsibility principles and strategy (either in a formal or informal way) should be in charge of the management function of the company [84]. The findings imply that implementation and evaluation of RI require different approaches and incentives depending on the nature of a company.

### 5.4. Limitations of the Research and Future Work

This work clearly has some limitations. First, various sectors and sizes of companies that were involved in the study represent different approaches and needs. Thus, the current state of the art may not be fully representative for every sector and company. Therefore, given a relatively small sample size, caution must be exercised in terms of generalisation. Second, for large companies, the interviews were conducted with a maximum two people per company, mainly with top level R&D, innovation or CSR/CS personnel. The SATORI project was constrained by limited time and resources. Therefore, it is recommended that further research should be carried out in the following areas: sector specific research on RI implementation and evaluation practices; and in-depth case studies of various companies involving interviews or survey with different units and departments to investigate the level of RI implementation, coherence and, ultimately a strategic approach to RI. Third, the field of responsible innovation has grown significantly since the interviews for this study were conducted. Therefore, it would be interesting to investigate whether and how perceptions and practices of businesses have changed over time.

### 6. Conclusions

This study contributes to knowledge about the implementation of responsible innovation (RI) in the business context combining insights from corporate social responsibility

(CSR) and corporate sustainability, ethics and innovation management of new and emerging technologies. To see how companies translate RI concept into practice, 24 interviews with companies and business experts were conducted within the SATORI project. The interviews illustrate companies' perceptions of RI, its role in their strategies and practices, evaluation and control approaches and methods.

The results show that RI is perceived as part of a broader CSR and CS framework. A variety of perspectives about RI in the business context are expressed; however, five main themes of responsibility related to innovation emerge from the analysis, namely: environmental responsibility, anticipation and reflection, social responsibility, stakeholder (both internal and external) and ethics. Moreover, the interviews suggest that RI in the business context is guided by four main principles, i.e., social responsibility, environmental impacts, professional integrity, and implications for health and/or safety. The vast majority of participants demonstrate their commitment to responsibility, ethics and sustainability. However, this commitment requires specific actions in terms of planning, implementation and evaluation and control. The interviews indicate that evaluation and control of innovation can be generally divided into three categories (1) assessment; (2) guidance; and (3) dissemination and awareness raising. The first two evaluation approaches are oriented towards evaluation of projects and practices on the one hand, and professional conduct on the other hand. The third category involves engagement in networks and trainings. Generally, companies evaluate and control their innovation activities using CSR and CS assessment tools, such as legal compliance assessment, impact assessment (IA), ethics assessment (EA), and safety assessment. Moreover, their evaluation and control are guided by various external and internal guidelines, codes of conduct and standards. The interviewees also emphasise the role of building a culture of responsibility within a company (employees) and its eco-system (business partners, supply chain). Training and multi-stakeholder and sectoral initiatives serve this purpose.

The interviews clearly confirm the theoretical confusion regarding the concepts of corporate responsibility, CSR, CS, ethics and RI. Therefore, this study indicates that three challenges need to be overcome to ensure effective application of responsible innovation in the business context. First, the definitions of CSR, CS and RRI and the relationship between these concepts should be clarified. This could be done by mapping shared meanings and relation between CSR, CS and RI tools, standards and indicators. Second, established indicators of RI in the business context could help in overcoming the conceptual confusion and operationalising and measuring responsibility (societal, ethical, environmental etc.). Such indicators are critical for implementation of RI and any kind of evaluation and assessment method. The RI indicators should be linked with well-known and already operationalised CSR and CS indicators and commonly used CSR and CS standards, such as ISO 26,000, Social Accountability 8000, OHSAS, ISO 14,001 and EMAS. At the same time, RI aspects could enrich CSR and CS tools and approaches, and in particular companies that focus on technological innovation would benefit from the concept of RI. General CSR and CS tools may not be well suited for tech companies and capturing innovation complexities. By bridging CSR, CS and RI, companies may develop more strategic innovation management through a cooperation and communication between technology developers and CSR/CS/ethics and human rights actors. Indicators are crucial for companies to measure their performance and impact, but also for customers, policy-makers and broader society to understand impacts that companies have on our lives. The RI quantitative indicators should be connected with KPIs, taking into consideration specificities of various sectors and companies. Such an evaluation and reporting framework should be multi-layered, providing general principles applicable to all types of actor as well as specific provisions suitable for different types and categories of actor (e.g., branches of industry, SMEs, large corporations). Third, currently research ethics is applied by companies mainly in the medical field. Companies outside the medical field, such as information technology (IT) and electronics, could also benefit from a paradigm of research ethics, particularly one type of practice, assessment of innovation e.g., via expert committees. However, there is a need

for methods that would help tech companies incorporating ethics, particularly for new and emerging technologies. Such methods could include eTA, eIA, anticipatory technology ethics (ATE), value-sensitive design (VSD), privacy for design, socially responsible design (SRD), eco-design, ethics by design etc.

A holistic and strategic approach for responsible innovation management, including evaluation and control, is needed to ensure that RI is implemented in a meaningful way, so that RI does not serve only marketing purposes, but helps companies to realise competitive opportunities while also leading to positive economic, societal and environmental impacts.

**Funding:** The research leading to these results received funding from the European Community's Seventh Framework Programme (FP7/2007–2013) under grant agreement No. 612231 (SATORI).

**Institutional Review Board Statement:** The research proposal, entitled: 'EU FP7 Science and Society project: SATORI', numbered BCE 15059, was approved for ethical conduct of scientific research by the Ethics Committee of the Faculty of Behavioral, Management and Social Sciences of the University of Twente. The empirical research was part of ethical assessment of the H2020 SATORI project.

**Informed Consent Statement:** Informed consent was obtained from all subjects involved in the study.

**Data Availability Statement:** The data are not publicly available due to confidentiality of research participants.

**Acknowledgments:** The author would like to acknowledge the contribution of all project's participants and all project's activities to the ideas that underpin this paper, particularly the authors of the SATORI report Gurzawska, A.; Cardone, R.; Porcari, A.; Mantovani, E.; Brey, P., 2015. SATORI Deliverable 1.1: Ethical Assessment of R&I: A Comparative Analysis; Annex 3h: Ethics Assessment in Different Types of Organizations: Industry, SATORI Project; and Shelley-Egan, C., Brey, P., Rodrigues, R., Douglas, D., Gurzawska, A., Bitsch, L. and Wadhwa, K., 2016. Ethical assessment of research and innovation: a comparative analysis of practices and institutions in the EU and selected other countries.

**Conflicts of Interest:** The author declares no conflict of interest.

## Appendix A. Interview Template

*Appendix A.1. PART A*

Appendix A.1.1. A. Interview Questions [Both Companies and Industry Experts and Organisations That Represent Industry]

1) Are you familiar with the concept of responsible innovation? How would you define responsible innovation?
2) (Questions about the way in which ethics assessment of research and/or innovation in performed)
   a) Can you describe what kind of ethical assessment your organisation does and what is its goal?
   b) And what is assessed: e.g., research proposals, research programs, policies, research results, technological innovations, behaviours of scientists and/ or innovators, etc.
   c) Who are the users (consumers) of the assessments?
   d) What kind of committee(s) or persons do the ethics assessment?
      i) What is their expertise?
      ii) How were they are chosen for this task?
      iii) Is there any consultation of stakeholders or of the public?
   e) Can you say which ethical values, principles or directives are used in ethical assessment in your organisation? For example, integrity, protection of human beings, promotion of the social good, informed consent, beneficence, justice, protection of the environment?
      i) Is there a shared framework of such values and principles or do individual assessors (also) bring their own values and principles to the table?

f) Which, if any, are the most important other organisations that you interact with in relation to ethics assessment? These may be organisations that have input into your assessments, regulate the way your organisation does assessments, are clients of your assessments, or that otherwise function as stakeholders.

g) Can you say how ethical assessment by your organisation is used and what its impact is?

    i) Are your recommendations binding or non-binding?

    ii) Are they generally followed; if not, how frequently are they followed, and what are the reasons that people or organisations have for not following them?

    iii) Is there any monitoring of compliance with your recommendations? If not, why not?

h) if you have performed any evaluations or assessments of the impact of ethics assessment as performed by your organisation,

    i) what have you found the impact to be?

    ii) where does ethics assessment function as desired, and where is it found wanting?

3) How would you assess the relative influence or importance of ethics assessment on research and innovation as compared to other forms of assessment, generally, and specifically within your company?

4) How would you describe the most important ethical problems in research and innovation that are assessed by your organisation?

a) Can ethical assessment performed by your organisation help solve these problems?

b) If not, what else is needed to solve them?

5) Are there weaknesses or problems in how ethical assessment takes place at your organisation? If so, can you please elaborate on their nature?

a) What actions are currently being taken or planned to improve ethical assessment?

b) What needs to change within or outside your organisation to make further improvements possible?

c) Do you think these problems might be addressed through capacity building and training activities? If yes, what kinds of needs should these activities address?

6) Do you think it would be is desirable to have a shared European approach for ethics assessment of research and innovation, with shared standards, procedures, and protocols for all European countries, and all organisations that engage in ethics assessment?

a) Do you believe it is possible?

b) What would be the obstacles to such an approach? What would be the benefits?

c) Would it be desirable for such an approach to have shared ethical values and principles, or only protocols and procedures?

d) If you are not sure if a shared approach for all types of organisation is desirable or feasible, do you think that it would be desirable for organisations of your type alone, that is, would you be interested in more shared standards and approaches with similar organisations in European member states?

Appendix A.1.2. Additional Questions for Companies

1) Is your company subjected to the new EU Directive on disclosure of non-financial and diversity information by large companies and groups?

2) If so, how do you approach the following disclosure of information: on environmental matters, social aspects, respect for human rights, anti-corruption and bribery issues?

3) Do you make any connection between these issues and (a) your CSR policies, and (b) ethical assessment of your R&D activities?

4) How can, in your opinion, ethical practices in R&D in industry best be improved? By what regulatory or self-regulatory tools?

5) What laws and regulations for corporate social responsibility and ethical research and innovation are you subjected to?
6) Do you have to file, under national legislation(s), any social and environmental impact statement and do these have any relation to ethical issues in research and innovation in your company?

Appendix A.1.3. Additional Questions for Industry Experts and Organisations That Represent Industry

1) How do your industry collaborators/members generally construe the relation between CSR policies and the assessment of ethical issues in research and innovation? Are they integrated activities or separate?
2) How, to your knowledge, is ethical assessment of R&I generally approached in the companies you collaborate with/represent?
    a) Are there big differences between different sectors (e.g., pharmaceutics, IT, agriculture, electronics, etc.)? Are there big differences between SMEs and large corporations? If so, which?
3) How do you expect the new EU Directive on disclosure of non-financial and diversity information by large companies and groups will affect EU companies, particularly their activities in research and innovation and their ethical assessment?
4) How can, in your opinion, ethical practices in R&D in industry best be improved? By what regulatory or self-regulatory tools?

*Appendix A.2. PART B*

Appendix A.2.1. B. Additional Factual Questions [Both Companies and Experts and Organisations That Represent Industry]

1) What is the full name of the organisation (in original language and in English, if available), and what is the name of the unit that engages in ethics assessment, if it is different? What is the website address?
2) Does the organisation have any policies or assessment procedures for the following, and if so, how are they used and how is compliance monitored, if at all?
    a) scientific integrity (avoiding scientific misconduct, such as fraud, data falsification, plagiarism, etc.)
    b) professional integrity (especially for innovators/engineers) (rules and principles for interacting with clients, employers, and other stakeholders, avoiding conflicts of interest, honesty, responsibilities to the environment, to general welfare, etc.)
    c) human subjects research (including special provisions for children and individuals who lack full autonomy)
    d) treatment of animal in experiments
    e) dealing with risks and anticipating social and environmental impacts, including
        i) implications for individual and civil rights, specifically:
            - freedom
            - non-discrimination and equality (are any specific groups mentioned, e.g., women, minorities, disabled, etc.)
            - autonomy
            - privacy
            - bodily integrity
            - human dignity
        ii) implications for (distributive) justice
        iii) implications for health and safety
        iv) implications for the environment
        v) implications for quality of life

<div style="text-align: right">

vi)    dual use (the possibility of military use of research and innovations)

</div>

f)    outsourcing of research and/or innovation to developing countries which may have lower ethics and/or social/environmental standards than the country in which the outsourcing agent is located.

3)    Does the organisation have any methods or procedures for assessing the impact of ethics assessment as performed by the organisation? Please state what they are.

Appendix A.2.2. Additional Factual Questions for Companies

1)    What is the company's policy, if any, for corporate social responsibility (CSR), and the units and personnel who are involved in it, and their relation to the rest of the organization?

To what extent does the CSR policy also cover ethical issues in research and innovation?

2)    Are there separate policies, units and personnel for the ethical assessment of research and innovation?

3)    Is the company's research and/or innovation assessed by any external ethics assessment bodies (for example, research ethics committees)?

4)    Does the company address ethical issues (such as the ones mentioned earlier in the interview) in its annual reports, and do these include ethical issues in research and innovation?

Appendix A.2.3. Additional Factual Questions for Industry Experts and Organisation That Represents Industry

1)    What is your role in ethical assessment of research and innovation, if any?

2)    Are you involved in setting professional standards for your constituents, lobbying government with respect to CSR or ethics standards and legislation, or other activities?

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
