# Peer review of "Responsible Innovation in Business: Perceptions, Evaluation Practices and Lessons Learnt"

_sustainability, doi:10.3390/su13041826_

Round 1
Reviewer 1 Report
Originality - The topic of the article is not new, but the current coronavirus pandemic makes it particularly relevant again. Several articles on the topic have been published e.g. in the framework of the Satori project, which is also the basis of this article. The interviews were conducted between September 2014 and May 2015, so, unfortunately, this information is no longer up-to-date, and in 5-6 years, both CSR and the ethics institution system in companies may have changed significantly. At the same time, the article provides useful information on the practical concept of responsible innovation.
Literature review - The literature review is a good summary of the concepts of CSR, CS and responsible innovation, but it would have been worthwhile to point out which definition the author considers most relevant.
Research design - The research questions of the study were relevant: how companies perceive responsibility in the context of their innovation practices and how they implement this responsibility? The research design met the research questions.
Method - The research method fits the research question. 24 interviews were carried out in person and via phone and Skype. In addition, desk research was conducted.
Sample - It is not clear how the sampled companies and experts were selected.
Results - Some of the interview questions were not answered in the article. e.g. How would you describe the most important ethical problems in research and innovation that are assessed by your organization? Are there weaknesses or problems in how ethical assessment takes place at your organization? Do you think it would be desirable to have a shared European approach to ethics assessment of research and innovation?
The article shows well what confusion there is about responsible innovation among companies, both in terms of definitions and practical implementation. It would be worthwhile to refer to studies in which companies can see good examples from which they can learn.
Formal proposals
References: The year is sometimes in parentheses, sometimes not. It should be standardized according to formal requirements.
There is no Crossref in the article.
Author Response
Dear Reviewer,
I am deeply grateful to the reviewer for taking the time to provide valuable comments and suggestions. Well-structured review with suggestions on specific additions and improvement was a great help in improving the manuscript. I revised the manuscript in accordance with your advice. Below I provide responses to your comments and I describe changes that were made:
Originality - The topic of the article is not new, but the current coronavirus pandemic makes it particularly relevant again. Several articles on the topic have been published e.g. in the framework of the Satori project, which is also the basis of this article. The interviews were conducted between September 2014 and May 2015, so, unfortunately, this information is no longer up-to-date, and in 5-6 years, both CSR and the ethics institution system in companies may have changed significantly. At the same time, the article provides useful information on the practical concept of responsible innovation.
The following remark was added in the limitation section: Last, the field of responsible innovation has grown significantly since the interviews for this study were conducted. Therefore it would be interesting to investigate whether and how perceptions and practices of businesses have changed over time.
Literature review - The literature review is a good summary of the concepts of CSR, CS and responsible innovation, but it would have been worthwhile to point out which definition the author considers most relevant.
The author added the following explanation: This study perceives corporate responsibility as a business strategy, where responsibility is designed to create business value and positive societal and environmental change and is managed in a systematic and intentional way (Husted and Allen 2010; Porter and Kramer 2006). Therefore social responsibility is embedded in a day-to-day business culture and operations (McElhaney 2009).
Research design - The research questions of the study were relevant: how companies perceive responsibility in the context of their innovation practices and how they implement this responsibility? The research design met the research questions.
Method - The research method fits the research question. 24 interviews were carried out in person and via phone and Skype. In addition, desk research was conducted.
Sample - It is not clear how the sampled companies and experts were selected.
The following explanation was added in the article: The sampled companies and experts were selected following the overall project’s methodology aiming at a mapping and comparative analysis the ethics assessment landscape for R&I in the EU, where countries were used as the main structuring principle for data collection. The sampled companies and experts represent both large companies and SMEs from various parts of EU, featuring different institutional and cultural arrangements.
Furthermore, an additional table (Table 1) was added providing an overview of companies’ interviewees (position, sector, major activity, size).
Moreover, this study was part of a broader research a comparative analysis of ethics assessment in the scientific fields, organisations and countries investigated. Over 230 interviews with representatives of organisations that engage in ethics assessment and guidance, and experts in the field. This broader study consists of detailed studies of ethics assessment in different scientific fields, types of organisations and countries, in addition to reports on major principles, issues and approaches in ethics assessment. The study on practices of industry was one part of this study. The author made a reference to a document explaining the project’s methodology.
Results - Some of the interview questions were not answered in the article. e.g. How would you describe the most important ethical problems in research and innovation that are assessed by your organization? Are there weaknesses or problems in how ethical assessment takes place at your organization? Do you think it would be desirable to have a shared European approach to ethics assessment of research and innovation?
First question was hardly answered, particularly its “ethical problems” dimension. Responses mainly focused on economic and organisational challenges rather than ethical per se. Clearly, terms like corporate responsibility, sustainability and ethics are mixed if not confused or misunderstood.
Second question, answers to this questions were presented following the understanding of the interviewee. Interviewees refer to “ethics assessment” rather in a combined form of any assessment related to corporate responsibility, not necessary ethics. Only companies from the pharma sector refer to ethics, mainly bioethics related to research with human subject and clinical trials when ethics approval from the ethics board is required. The interviewees did not point out any specific weaknesses. Other sectors do not refer to ethics assessment therefore they did not point out any weaknesses.
Third question about a shared European approach to ethics assessment of research and innovation raises a broad discussion about unification of RRI approaches among research and innovation organisation. However in the context of business this discussion is the tip of the iceberg, a broader discussion about companies as subjects of international law and conflict between hard low and soft law/self-regulation (e.g. The UN treaty on transnational corporations and human rights). This point is discussed in the original study by the SATORI project. However, the author did not want to lose the focus of this study and therefore focused on the main outcomes of the interviews. The interviewees emphasise variety of businesses, and variety of guidelines, frameworks and standards they follow. Thus, the article indirectly discusses the issue.
The article shows well what confusion there is about responsible innovation among companies, both in terms of definitions and practical implementation. It would be worthwhile to refer to studies in which companies can see good examples from which they can learn.
The following text was added in the discussion section: One example is Responsible Innovation COMPASS self-check tool developed with intention to help SMEs determine to what extent their practices align with RI principles, how to improve their innovation processes and outcomes, and how they compare to other companies [84]. The MoRRI project developed a list if RRI indicators for adequate measurement of responsibility in research and innovation, which could serve as KPIs [85]. Other initiatives provide lessons-learnt through pilot studies engaging companies [86] and co-creation of good practices through workshops and community networks [87, 88].
The following reference points were added:
Responsible Innovation COMPASS Self-assessment Tool, https://innovation-compass.eu/self-check/, accessed in January 2021.
MoRRI EU-funded project, http://morri-project.eu/, accessed in January 2021.
PRISMA EU-funded project, https://www.rri-prisma.eu/pilot-projects/, accessed in January 2021.
LIVING INNOVATION EU-funded project, https://www.living-innovation.net/explore, accessed in January 2021.
The Prince’s Responsible Business Network, (2020), Responsible Innovation Framework, https://www.bitc.org.uk/wp-content/uploads/2020/09/bitc-digital-report-responsibleinnovationframework-sep20.pdf , accessed in January 2021.
Formal proposals
References: The year is sometimes in parentheses, sometimes not. It should be standardized according to formal requirements.
Adapted.
There is no Crossref in the article.
Reviewer 2 Report
Dear author,
I really enjoyed to read the manuscript. It is well documented, de literature review and context is deeply analyzed. The research has some interesting conclusions, and the review of the scientific literature itself has a certain value, and the manuscript can published based on that. That would be my first recommendation. Namely, to change the title and the type of the manuscript, and reedit it as a kind of review article.
The major weakness of the manuscript consists of the small number of interviews. Also, among these few interviews, we have a mixture of business practitioners, NGO people, scientists and others. That dilutes the value of the conclusions. I understand the reason why the sample is so heterogeneous, at least in the case of interviewees, but still, there are several fields of industries and activities in which innovation is important and are not covered by the research. In spite of this fact you make conclusions that lets the reader to assume that they are generally valid. Furthermore, the article is based on data collected 7 years ago! Some of the industries analyzed are extremely innovative, so it might be a change in the opinion of the interviewees since than. My second proposal is to change the research type in vertical research, and I mean by that to repeat the interviews (maybe in a shorter form) and to compare the evolution of the views. I think, in very short time you might have extremely valuable data.
As overall consideration the manuscript is good, it is easy to read it. I hope it will be published soon, but it needs in my opinion some minor improvements.
Good luck!
24 interviews interviews were conducted between September 2014 and May 2015.
Author Response
Dear Reviewer,
I am deeply grateful to the reviewer for taking the time to provide valuable comments and suggestions. Well-structured review with suggestions on improvement was a great help in improving the manuscript. I revised the manuscript to the most feasible level.
First, the author agrees with the limitations of the sample, therefore a provided further explanation about the sample as well as its limitations.
The following explanation was added in the article: The sampled companies and experts were selected following the overall project’s methodology aiming at a mapping and comparative analysis the ethics assessment landscape for R&I in the EU, where countries were used as the main structuring principle for data collection. The sampled companies and experts represent both large companies and SMEs from various parts of EU, featuring different institutional and cultural arrangements.
Additional information about the sample was added (Table 1) specifying interviewees' positions, sector, major activity, and size of the company.
Moreover, this study was part of a broader research a comparative analysis of ethics assessment in the scientific fields, organisations and countries investigated. Over 230 interviews with representatives of organisations that engage in ethics assessment and guidance, and experts in the field. This broader study consists of detailed studies of ethics assessment in different scientific fields, types of organisations and countries, in addition to reports on major principles, issues and approaches in ethics assessment. The study on practices of industry was one part of this study. The author made a reference to a document explaining the project’s methodology.
The text was changed in the following way:
This work clearly has some limitations. First, various sectors and sizes of companies that were involved in the study representing different approaches and needs. Thus, current state-of-the-art may not be fully representative for every sector and company. Therefore, given a relatively small sample size, caution must be exercised in terms of generalisation. Second, for large companies, the interviews were conducted with a maximum two people per company, mainly with top level R&D, innovation or CSR/CS personnel. The SATORI project was constrained by limited time and resources. Therefore, it is recommended that further research should be carried out in the following areas: sector specific research on RI implementation and evaluation practices; and in-depth case studies of various companies involving interviews or survey with different units and departments to investigate the level of RI implementation, coherence and, ultimately a strategic approach to RI. Third, the field of responsible innovation has grown significantly since the interviews for this study were conducted. Therefore it would be interesting to investigate whether and how perceptions and practices of businesses have changed over time.
The suggestion about reopening the interview process is definitely interesting and it would be valuable to see how much these perceptions and perspectives have changed. Unfortunately, due to lack of resources such study is challenging. This study was part of the large EU-funded project, which now is finished. Moreover, I double-checked and a number of interviewees already changed companies and their positions. It might affect the interviews and as a result they would not be relevant for a specific company anymore.
Beside limitations, the author's intention is to share valuable outcomes of the study, particularly about the language and concept confusion (which is still relevant) and point out to the need of a systematic and strategic approach to responsibility, also unfortunately still in majority valid. The author hopes that the strengths of this study overcomes its flows, and therefore a revised version would be accepted for publication.
Kind regards
Reviewer 3 Report
In my opinion, the article is an interesting, but popularly scientific text. Purpose of the work "research aims to the extent to which similarities and differences exist in the use of frameworks and procedures. This paper discusses theoretical and practical implications of discrepancies in definitions of responsibility, sustainability and ethics, language used, differences between .... "is a very general purpose.
I find the article interesting. But I think the research part should be completely improved. Research not conducted correctly. There is no valid statistics at work. Besides, the analysis of secondary research is not sufficient for publication in the journal "Sustainability".
Maybe the article is appropriate in the sciences - like Philosophy, but from the point of view of management sciences it is too general.
Author Response
Dear Reviewer,
I am deeply grateful to the reviewer for taking the time to provide valuable comments and suggestions. Well-structured review with suggestions on improvement was a great help in improving the manuscript. I revised the manuscript.
First, the author agrees with the limitations of the sample, therefore I provided further explanation about the sample as well as its limitations.
The following explanation was added in the article: The sampled companies and experts were selected following the overall project’s methodology aiming at a mapping and comparative analysis the ethics assessment landscape for R&I in the EU, where countries were used as the main structuring principle for data collection. The sampled companies and experts represent both large companies and SMEs from various parts of EU, featuring different institutional and cultural arrangements.
Additional information about the sample was added (Table 1) specifying interviewees' positions, sector, major activity, and size of the company.
Moreover, this study was part of a broader research a comparative analysis of ethics assessment in the scientific fields, organisations and countries investigated. Over 230 interviews with representatives of organisations that engage in ethics assessment and guidance, and experts in the field. This broader study consists of detailed studies of ethics assessment in different scientific fields, types of organisations and countries, in addition to reports on major principles, issues and approaches in ethics assessment. The study on practices of industry was one part of this study. The author made a reference to a document explaining the project’s methodology.
The text was changed in the following way:
This work clearly has some limitations. First, various sectors and sizes of companies that were involved in the study representing different approaches and needs. Thus, current state-of-the-art may not be fully representative for every sector and company. Therefore, given a relatively small sample size, caution must be exercised in terms of generalisation. Second, for large companies, the interviews were conducted with a maximum two people per company, mainly with top level R&D, innovation or CSR/CS personnel. The SATORI project was constrained by limited time and resources. Therefore, it is recommended that further research should be carried out in the following areas: sector specific research on RI implementation and evaluation practices; and in-depth case studies of various companies involving interviews or survey with different units and departments to investigate the level of RI implementation, coherence and, ultimately a strategic approach to RI. Third, the field of responsible innovation has grown significantly since the interviews for this study were conducted. Therefore it would be interesting to investigate whether and how perceptions and practices of businesses have changed over time.
Sustainability Journal is described as "an international, cross-disciplinary, scholarly (...) journal of environmental, cultural, economic, and social sustainability of human beings. It provides an advanced forum for studies related to sustainability and sustainable development". This article is in line with the journal's objectives and topics covered. One example is Special Issue "Responsible Research and Innovation (RRI) in Industry", which covers similar areas of research. Moreover, it covers such topics as barriers, obstacles, incentives, bridging perspectives, topics quite similar in their scope to this article.
Beside limitations, the author's intention is to share valuable outcomes of the study, particularly about the language and concept confusion and point out to the need of a systematic and strategic approach to responsibility. The author hopes that the strengths of this study overcome its flows, and therefore a revised version will be accepted for publication.
Kind regards
Round 2
Reviewer 2 Report
Dear Author,
Thank you for your answer, indeed the article is much better now. Still, I don't know how could be upgraded... I'm still concerned about the lack of the consistency of the sample (beside the size of the sample), and the date of the collected data... It seems a little bit outdated, however the conclusions are probably still valid. The conclusions are too general.
Reviewer 3 Report
Thanks for the author's answers. As I wrote before, the article is interesting and nice to read. In my opinion, the article is very general from a management point of view.
However, I appreciate the author's commitment and the changes made.
Congratulations!
This manuscript is a resubmission of an earlier submission. The following is a list of the peer review reports and author responses from that submission.